

# Study on the correlation between multiple object tracking ability and eye-tracking characteristics in sports decision making among basketball players

Qifeng Gou[1,2,*] and Sunnan Li[2,*]

[1] College of Physical Education, The Northwest Normal University, Lanzhou, China
[2] College of P.E. and Sports, Beijing Normal University, Beijing, China
[*] These authors contributed equally to this work.

## ABSTRACT

**Background**. Variations in attention and search strategies can partially explain the differences in players' decision-making capabilities. This study aims to explore the correlation between basketball players' multiple object tracking (MOT) ability and visual search in sports decision-making tasks using the expert-novice paradigm.

**Methods**. Forty-eight college students were selected and divided into two groups: the expert group (24 basketball players) and the novice group (24 general college students). Using different numbers and speeds in MOT tasks, the Eyelink eye tracker was employed to record the visual search process during sports decision-making tasks. The correlation coefficient between MOT accuracy and eye-tracking indicators was then analyzed.

**Results**. (1) The accuracy of MOT in both the expert and novice groups was negatively correlated with average fixation time and average fixation frequency. The correlation between MOT accuracy and average fixation time for passing intuition and passing cognition in the expert group was stronger than that in the novice group. (2) When tracking three–six targets at a speed of 10°/s, a strong correlation was observed between the MOT accuracy in both groups and the eye movement indicators of the sports decision-making task. (3) The correlation between MOT accuracy in the expert group and the proportion of fixation time in irrelevant areas during intuitive decision-making was lower than in the novice group. Similarly, the correlation with the proportion of fixation frequency in irrelevant areas during cognitive decision-making was also lower in the expert group compared to the novice group.

**Conclusions**. The multiple object tracking ability of basketball players is negatively correlated with fixation time and fixation frequency in sports decision-making tasks, and positively correlated with fixation time during passing decisions.

Corresponding author
Qifeng Gou,
202031070018@mail.bnu.edu.cn

## INTRODUCTION

The process through which players formulate action plans based on acquired information in complex sports situations, while weighing the pros and cons in a limited time frame, is referred to as sports decision-making (*Ren, 2019*). The first step in motor decision-making

is perception, where neurons transmit information to the central nervous system to facilitate decision-making (*Bahdur, 2015*). Over 80% of brain information is derived from visual attention (*Sui, Gao & Xiang, 2018*). High-quality attention enables players to selectively focus, recognize, and interpret visual information more effectively, providing the necessary information for further decision-making and response (*Liao, Zhang & Ge, 2004*). Variations in attention and search strategies can partially explain the differences in players' decision-making capabilities (*Chu & Wang, 2024*).

Multiple object tracking (MOT), also known as multi-target tracking, involves the ability to track and focus on the movement of several objects simultaneously (*Pylyshyn & Storm, 1988*). MOT integrates the core elements of attention: selectivity, limited capacity, and subjective effort, reflecting aspects such as selective attention, attentional allocation, and sustained attention (*Faubert, 2013*). This paradigm shifts the focus of attention research from static attention to dynamic attention, significantly enhancing the ecological validity of attention studies. In team ball sports, players must simultaneously monitor multiple teammates, opponents, and positional changes. The ability to track multiple targets is crucial for success in such sports (*Faubert & Sidebottom, 2012*).

In modern basketball, players perform numerous offensive and defensive transitions. To make accurate predictions and execute corresponding technical movements, players must rapidly identify the most critical information in fast-paced situations. Consequently, enhancing the attention capacity of basketball players to improve decision-making accuracy has become a focal point for coaches and sports researchers worldwide (*Jin et al., 2020a*). For tactical coordination, dribbling players need to identify passing opportunities from teammates, reflecting the selective function of attention (*Mao, 2010*). Defenders must shift their focus from preventing direct passes to blocking behind passes to intercept the opponent's pass. Slow transfer of focus may lead to failure in defense. A basketball guard must also exhibit broad attention to the entire court, sometimes engaging in man-to-man defense, requiring stability in attention. While dribbling, the player must monitor the court and make tactical decisions, requiring attention allocation for both dribbling and passing. In the penalty area, defenders must quickly adjust their focus to replenish the defense when the opponent breaks through (*Wu, 2005*). If basketball players lack effective attention allocation and shifting abilities, they are unable to fully utilize their technical and tactical skills (*Yao, 2020*). Thus, basketball requires comprehensive attention abilities, including multiple object tracking, broad external attention, narrow internal attention, and efficient attention allocation.

Sports decision-making is a critical indicator of athletic performance, as players must rapidly process dynamic changes on the field to make informed choices. There are two types of decision-making in sports: cognitive decision-making and intuitive decision-making. Cognitive decision-making resembles general decision-making processes, while intuitive decision-making involves quick judgments under time constraints (*Wang, 2005*). Most existing studies utilize static images as stimuli, but dynamic videos offer higher ecological validity, especially when a first-person perspective is used. A first-person perspective provides a close-up view of the objects within reach of the protagonist, helping the observer better predict the next action. This perspective influences individuals' understanding of the

environment, spatial structure, and predictions of the protagonist's subsequent activities (*Zhao, 2020*). This study uses first-person basketball video materials to enhance the ecological validity of the research. Effective visual attention in basketball players is not only fundamental for responding to the most relevant visual cues during the game but is also crucial for making accurate decisions and executing sports skills. Therefore, it is essential to explore the relationship between general task visual attention and visual attention in specific decision-making tasks. Based on this, the current study investigates the correlation between players' multiple object tracking abilities and their eye movement characteristics in decision-making tasks, analyzing visual search patterns from MOT performance and decision-making tasks.

Research hypothesis: players with strong MOT abilities may demonstrate enhanced visual information processing in sports decision-making tasks, with a negative correlation to fixation time and fixation frequency. Expert players are expected to show a stronger correlation between general task visual attention (MOT accuracy) and specialized decision-making task visual attention (eye movement indicators), and a lower correlation with irrelevant area attention allocation.

## MATERIALS & METHODS

### Participants

The study used G*Power 3.1.9.7 software to estimate the sample size, setting the effect size as biased $\eta^2 = 0.03$ (*Lakens, 2014*) and $\alpha = 0.05$. The calculation indicated that 40 participants (20 in each group) would achieve a statistical test power of 0.80. In practice, 48 participants (24 in each group) were selected (*Jin, Ge & Fan, 2023*). These participants were divided into an expert group and a novice group based on their basketball experience and skill level. The expert group was composed of players from the Northeast Division of the Chinese University Basketball League First Division (*Pylyshyn & Annan, 2006*), with a sports level of level one or above (including level one), an average age of $(21.20 \pm 2.12)$ years, and an average training period of $(9.10 \pm 2.14)$ years. The novice group was composed of 24 students from the basketball elective course at Northwest Normal University, with an average age of $(19.83 \pm 0.89)$ years, an average training period of $(1.60 \pm 0.49)$ years, and a weekly training time of $(1.20 \pm 0.15)$ hours, with no formal sports level. All participants were female. The experiment was approved by the Ethics Committee of Northwest Normal University, No. NWNU-20230301. Prior to the experiment, participants were informed of the purpose and procedures and provided written informed consent.

### Design

Experiment 1a utilizes a multiple object tracking task to examine tracking performance under different target numbers. A mixed experimental design of 2 (group: expert, novice) $\times 5$ (number of targets: two, three, four, five, six) was employed. Experiment 1b utilizes a multiple object tracking task to examine tracking performance at different target velocities. A mixed experimental design of 2 (group: expert, novice) $\times 3$ (target velocities: 5°/s, 10°/s, 15°/s) was employed. The accuracy algorithm is based on the percentage of each participant correctly selecting the target in all experimental trials.

Experiment 2 adopts a design of 2 (group: expert, novice) × 2 (decision type: intuition, cognition) × 3 (attack style: passing, shooting, breakthrough). The eye movement indicators for decision-making tasks include average fixation time, average fixation frequency, proportion of fixation time in different areas of interest, and proportion of fixation frequency in different areas of interest.

## Apparatus

Experiment 1 utilized a Lenovo laptop with a 15.6-inch display screen, a resolution of 1,920× 1,080 pixels, and a refresh rate of 60 Hz. A multiple object tracking experimental program was developed using Matlab R2020b software. The presentation of experimental materials and the key presses of subjects were automatically recorded by the program according to a pre-compiled control file.

Experiment 2 was conducted using the Eyelink eye-tracking data acquisition and analysis system developed by SR Research Ltd. (SR Research Ltd, 2023). A Lenovo laptop was used as the experimental subject, with a 15.6-inch display screen, a screen resolution of 1,920 × 1,080 pixels, and a refresh rate of 60 Hz. The Eyelink Portable Duo portable eye tracker was used to track the eye movement of the subjects with a sampling rate of 1,000 Hz.

## Stimuli

At the beginning of Experiment 1a, a "+" appeared in the center of the gray screen for 500 ms. Subsequently, 12 white spheres appeared on the screen, with 2, 3, 4, 5, or 6 spheres changing from white to blue, flashing three times to mark them as target objects. The remaining spheres, which did not change color, were non-target objects. The spheres then turned back to white and moved independently at a speed of 5°/s. During this period, occlusion may occur. After 8 s, the spheres stopped moving, and those that matched the target objects turned red. Participants were instructed to determine how many of the red spheres were target objects and press the corresponding number keys to proceed to the next trial.

For Experiment 1b, the procedure followed was similar, but the spheres moved at different speeds of 5°/s, 10°/s, and 15°/s. After 8 s, the spheres stopped, and the target objects turned red. Participants were then asked to identify the number of target objects in the red spheres and press the corresponding number key.

For Experiment 2, the offensive video stimuli were selected from the frontcourt offensive positions of the Women's Chinese Basketball Association (WCBA) league and were jointly determined by basketball coaches and players. Ten high-level female basketball players were recruited to recreate the game scenes on site. The players wore black and white jerseys, and the team members wore sports cameras (model: Insta360 One X) to capture the first-person perspective (*Ping, 2019*). After several practice runs, the official filming began, and the video quality was approved by the coach and filming team members.

The selection criteria for the video materials were: (1) the scene should depict a successful attack; (2) the number of players on screen should be sufficient (at least five offensive and defensive players visible simultaneously, following the guidelines of *Wang (2005)*; (3) the team members' movements should be clear.

The video materials underwent validity testing. Six high-level basketball players and six college students enrolled in basketball elective courses were selected to conduct validity testing. For videos with low discrimination, the duration was adjusted until the video effectively distinguished the decision-making agility and rationality between high-level players and general college students. The final video stimulus duration was set at 3,000 ms (*Wang, 2010*).

Experiment 2 included a total of 129 videos, consisting of nine practice videos and 120 formal experimental videos (40 each for passing, shooting, and breakthrough). The number of cognitive and intuitive decision-making tasks was equal (*Zhang, 2021*), ensuring consistent decision probabilities for the three offensive techniques.

## Division of interest areas

Areas of interest (AOIs) refer to the specific areas in the stimulus materials that researchers focus on, which are defined based on the experimental hypotheses and objectives. For video-based experiments, dynamic areas of interest were identified and drawn frame by frame with a mouse. The defined area of interest template was applied to all subjects. In consultation with two basketball coaches, one active Chinese Basketball Association (CBA) player, and one Eyelink engineer, the interest areas were categorized into key areas of interest, associated areas of interest, and irrelevant areas of interest (*Jin, 2020*). The key areas of interest included the player's ball-handling area and areas where key actions (such as passing, shooting, and breaking) are about to occur—these are crucial in the decision-making process. Associated areas of interest included potential passing targets, possible defensive positions, and teammate locations, which are relevant to decision-making but not immediate focus points. Irrelevant areas of interest included the boundaries of the venue, spectator stands, and other areas not directly related to the decision-making process. These areas were used for comparative analysis of players' attention allocation. See Fig. 1.

## Procedure

Experiment 1 was conducted in the laboratory, with participants testing on the same computer. Prior to the experiment, participants were familiarized with the button positions. Five pre-experiments were conducted to allow participants to familiarize themselves with the program before the formal trial. Experiment 1a consisted of 10 blocks, each with five trials corresponding to two, three, four, five and six targets, for a total of 50 trials. A 10-second white screen break was provided between blocks to alleviate eye fatigue, and the experimental sequence was balanced within subjects. The entire experiment lasted approximately 17 min. To prevent participants from prioritizing reaction speed at the expense of accuracy, the experiment did not require rapid key presses. The process of Experiment 1a is shown in Fig. 2.

Experiment 1b also consisted of 10 blocks, with each block containing three trials corresponding to three different target speeds: 5°/s, 10°/s, and 15°/s, for a total of 30 trials. A 10-second white screen break was provided between each block to alleviate eye fatigue. The experimental sequence was balanced within subjects. The entire experiment took approximately 11 min.

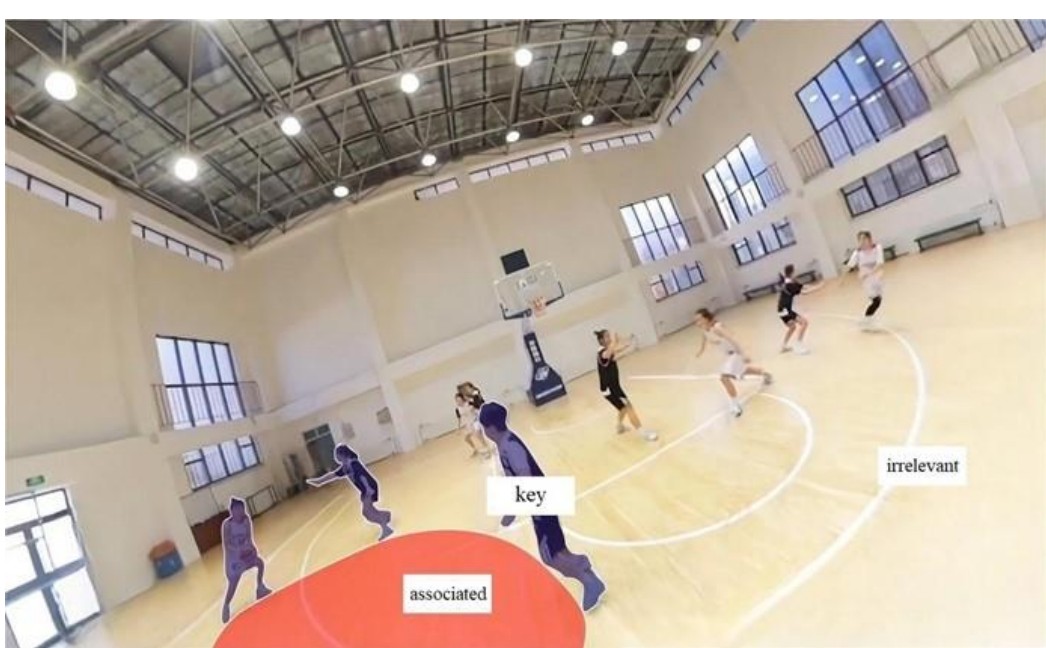

**Figure 1** Example of interest area editing.

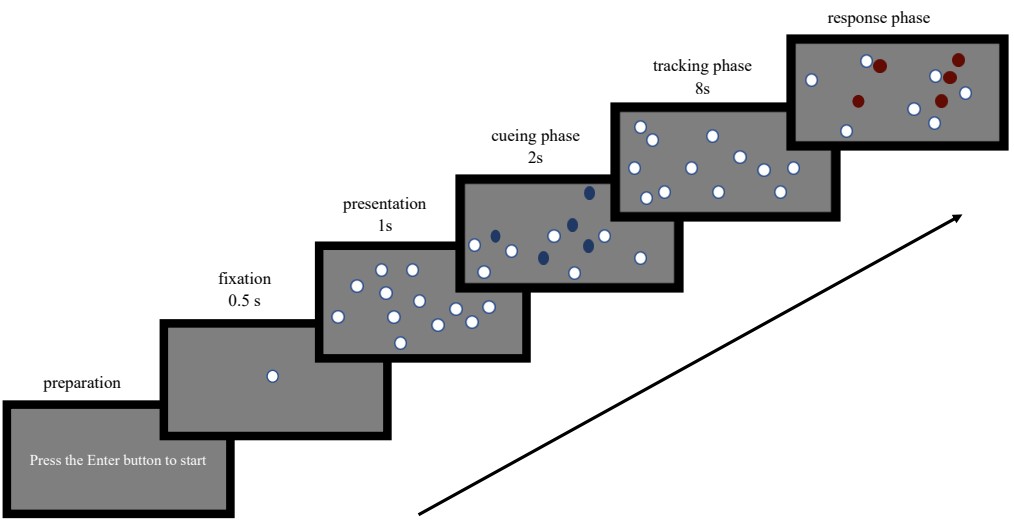

**Figure 2** Multiple object tracking task process (different number).

The Experiment 2 program was developed using Experiment Builder 2.3 software, with the stimulus material presented in the full-screen area. The presentation of stimuli and the key presses by participants were automatically recorded by the program with an accuracy of 1 ms. The process of Experiment 2 is shown in Fig. 3.

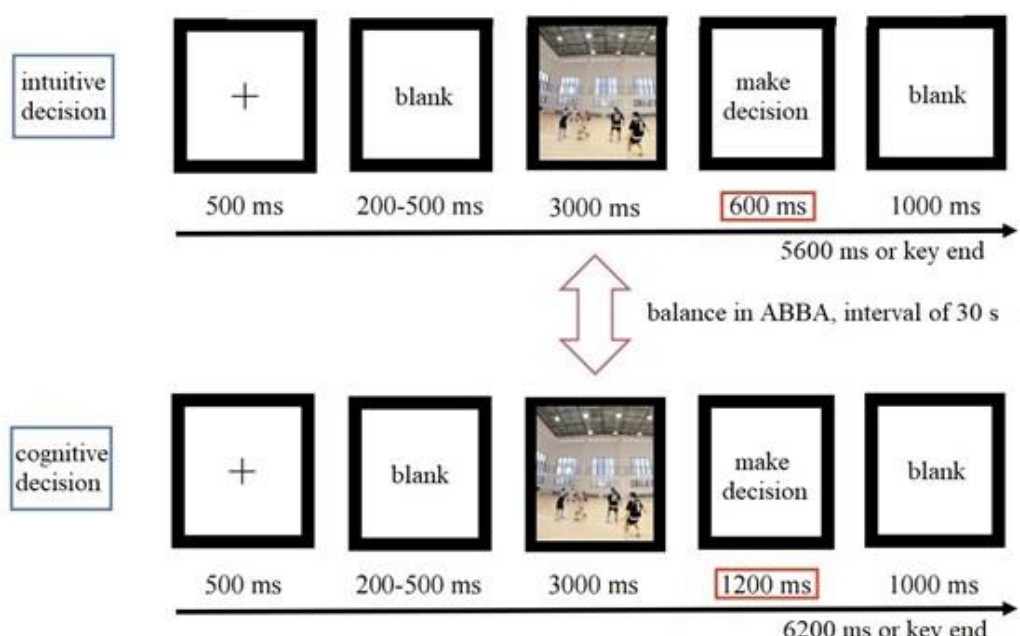

**Figure 3 Basic process of eye tracking experiment.**

## Statistical analysis

Experiment 1 collected data using Matlab R2020b software, while Experiment 2 used Experiment Builder 2.3 software for data collection and Data Viewer 4.2 for preprocessing. The collected data were then exported in EXCEL format. Following discussions with experts, the eye movement feature data were organized into the following categories: average fixation time, average fixation frequency, proportion of fixation time in the area of interest, and proportion of fixation frequency in the area of interest.

Correlation analysis was conducted using SPSS 29.0 software. The study employed the Pearson correlation coefficient to test for linear relationships, with a significance threshold of $P < 0.05$. The data follows a normal distribution using the Shapiro–Wilk normality test. The degree of correlation between variables was represented by the correlation coefficient (R), which ranges from −1 to 1. A positive correlation is indicated by a value greater than 0, while a negative correlation is indicated by a value less than 0. The absolute value of the correlation coefficient ($|R|$) was interpreted as follows: a low correlation for values between 0 and 0.30, a moderate correlation for values between 0.31 and 0.49, a high correlation for values between 0.50 and 0.69, a very high correlation for values between 0.70 and 0.89, and an approximately linear correlation for values between 0.90 and 1 (*Hopkins et al., 2009*). The *post-hoc* test employed the Bonferroni.

## RESULTS

### Correlation between MOT accuracy and average fixation time

The results indicated a moderate negative correlation ($|R| > 0.31$, $Ps > 0.05$) between tracking accuracy and the average fixation time for passing intuition and passing cognition in both the expert and novice groups when the number of MOT targets was two. When the number of MOT targets was three–four, tracking accuracy in both the expert and novice groups was highly negatively correlated with the average fixation time for passing intuition and passing cognition ($|R| > 0.69$, $Ps < 0.001$). When the number of MOT targets was five, tracking accuracy in both the expert and novice groups was highly negatively correlated with the average fixation time for passing intuition and passing cognition ($|R| > 0.50$, $Ps < 0.001$). When the number of MOT targets was six, tracking accuracy in the expert group was moderately negatively correlated with the average fixation time for passing intuition ($R = -0.491$, $P = 0.015$) and highly negatively correlated with the average fixation time for passing cognition ($R = -0.529$, $P = 0.008$). In the novice group, tracking accuracy was moderately negatively correlated with the average fixation time for both passing intuition and passing cognition ($|R| > 0.45$, $Ps < 0.05$).

When the MOT target speed was 5°/s, both the expert and novice groups showed a low negative correlation ($|R| < 0.30$, $Ps > 0.05$) between tracking accuracy and the average fixation time for passing intuition and passing cognition. At 10°/s, tracking accuracy in the expert group was highly negatively correlated with the average fixation time for both passing intuition ($R = -0.689$, $P < 0.001$) and passing cognition ($R = -0.783$, $P < 0.001$). In the novice group, tracking accuracy was moderately negatively correlated with the average fixation time for passing intuition ($R = 0.413$, $P = 0.045$) and highly negatively correlated with the average fixation time for passing cognition ($R = 0.742$, $P < 0.001$). At 15°/s, tracking accuracy in the expert group was weakly negatively correlated with the average fixation time for passing intuition ($R = -0.190$, $P = 0.374$) and moderately negatively correlated with the average fixation time for passing cognition ($R = -0.352$, $P = 0.092$). In the novice group, tracking accuracy showed weak negative correlations with the average fixation time for both passing intuition and passing cognition ($|R| < 0.30$, $Ps > 0.05$).

Overall, the correlation between MOT accuracy and average fixation time was negative in both the expert and novice groups, with the expert group showing stronger correlations between MOT accuracy and the average fixation time for both passing intuition and passing cognition than the novice group. See Table 1.

### Correlation between MOT accuracy and average fixation frequency

The results indicated a low negative correlation ($|R| < 0.30$, $Ps > 0.05$) between tracking accuracy and the average fixation frequency for passing intuition and passing cognition in both the expert and novice groups when the number of MOT targets was two. When the number of MOT targets was three–four, tracking accuracy in both the expert and novice groups was highly negatively correlated with the average fixation frequency for passing intuition and passing cognition ($|R| < 0.70$, $Ps < 0.01$).

**Table 1 Correlation between MOT accuracy and average fixation time.**

| MOT | Correlation | Expert | | Novice | |
|-----|-------------|--------|--------|--------|--------|
| | | Passing intuition | Passing cognition | Passing intuition | Passing cognition |
| 2 | R | −0.367 | −0.334 | −0.354 | −0.310 |
| | P | 0.077 | 0.110 | 0.089 | 0.140 |
| 3 | R | −0.757** | −0.806** | −0.708** | −0.735** |
| | P | <0.001 | <0.001 | <0.001 | <0.001 |
| 4 | R | −0.737** | −0.780** | −0.697** | −0.718** |
| | P | <0.001 | <0.001 | <0.001 | <0.001 |
| 5 | R | −0.638** | −0.657** | −0.604** | −0.560** |
| | P | <0.001 | <0.001 | <0.002 | 0.004 |
| 6 | R | −0.491* | −0.529** | −0.456* | −0.468* |
| | P | 0.015 | 0.008 | 0.025 | 0.021 |
| 5°/s | R | −0.219 | −0.204 | −0.191 | −0.139 |
| | P | 0.303 | 0.339 | 0.372 | 0.516 |
| 10°/s | R | −0.689** | −0.783** | −0.413* | −0.742** |
| | P | <0.001 | <0.001 | 0.045 | <0.001 |
| 15°/s | R | −0.190 | −0.352 | −0.062 | −0.253 |
| | P | 0.374 | 0.092 | 0.775 | 0.233 |

Notes.

*$P < 0.05$.

**$P < 0.01$.

When the number of MOT targets was five, tracking accuracy in the expert group was moderately negatively correlated with the average number of fixations for passing intuition ($R = -0.469$, $P = 0.021$), while in the novice group, the correlation was moderate but not statistically significant ($R = -0.320$, $P = 0.127$). The tracking accuracy of both the expert and novice groups was moderately negatively correlated with the average number of fixations for passing intuition ($|R| > 0.50$, $Ps < 0.01$).

When the number of MOT targets was six, a moderate negative correlation was found between tracking accuracy in both groups and the average number of fixations for passing intuition ($|R| > 0.31$, $Ps < 0.05$). The tracking accuracy of the expert group was moderately negatively correlated with the average fixation frequency for passing cognition ($R = -0.331$, $P = 0.114$), while the novice group showed a high negative correlation ($R = -0.502$, $P = 0.012$).

At an MOT target speed of 5°/s, a low negative correlation was found between the tracking accuracy of both the expert and novice groups and the average number of fixations for passing intuition ($|R| < 0.30$, $Ps > 0.05$). The expert group exhibited a moderate negative correlation with the average fixation frequency for passing cognition ($R = -0.316$, $P = 0.132$), while the novice group showed a weak negative correlation ($R = -0.303$, $P = 0.150$).

When the MOT target speed was 10°/s, tracking accuracy in the expert group was highly negatively correlated with the average number of fixations for both passing intuition ($R = -0.598$, $P = 0.002$) and passing cognition ($R = -0.708$, $P < 0.001$). In the novice

**Table 2   Correlation between MOT accuracy and average fixation frequency.**

| MOT | Correlation | Expert | | Novice | |
|---|---|---|---|---|---|
| | | Passing intuition | Passing cognition | Passing intuition | Passing cognition |
| 2 | R | −0.253 | −0.218 | −0.235 | −0.261 |
| | P | 0.232 | 0.307 | 0.268 | 0.218 |
| 3 | R | −0.688** | −0.694** | −0.519** | −0.671** |
| | P | <0.001 | <0.001 | 0.009 | <0.001 |
| 4 | R | −0.616** | −0.625** | −0.538** | −0.635** |
| | P | 0.001 | 0.001 | 0.007 | <0.001 |
| 5 | R | −0.469* | −0.526** | −0.320 | −0.523** |
| | P | 0.021 | 0.008 | 0.127 | 0.009 |
| 6 | R | −0.412* | −0.331 | −0.476* | −0.502* |
| | P | 0.046 | 0.114 | 0.019 | 0.012 |
| 5°/s | R | −0.277 | −0.316 | −0.227 | −0.303 |
| | P | 0.190 | 0.132 | 0.286 | 0.150 |
| 10°/s | R | −0.598** | −0.708** | −0.471* | −0.604** |
| | P | 0.002 | <0.001 | 0.020 | 0.002 |
| 15°/s | R | −0.332 | −0.287 | −0.338 | −0.375 |
| | P | 0.112 | 0.174 | 0.106 | 0.071 |

**Notes.**
*$P < 0.05$.
**$P < 0.01$.

group, the tracking accuracy was moderately negatively correlated with the average fixation frequency for passing intuition ($R = -0.471$, $P = 0.020$) and highly negatively correlated with the average fixation frequency for passing cognition ($R = -0.604$, $P = 0.002$).

At a target speed of 15°/s, a moderate negative correlation was found between tracking accuracy in both groups and the average number of fixations for passing intuition ($|R| > 0.31$, $Ps > 0.05$). The expert group showed a low negative correlation with the average fixation frequency for passing cognition ($R = -0.287$, $P = 0.174$), while the novice group exhibited a moderate negative correlation ($R = -0.375$, $P = 0.071$).

Overall, the correlation between MOT accuracy and the average number of fixations was negative in both the expert and novice groups. See Table 2.

## Correlation between MOT accuracy and intuitive decision area of interest gaze time ratio

The results indicated that when the number of MOT targets was three, tracking accuracy in the expert group was highly negatively correlated with the proportion of gaze time in the key and associated areas of passing intuition ($|R| > 0.50$, $Ps < 0.01$). In contrast, tracking accuracy in the novice group was highly negatively correlated with the proportion of gaze time in the key area of passing intuition ($R = -0.570$, $P = 0.004$).

When the number of MOT targets was four, the tracking accuracy of the expert group remained highly negatively correlated with the proportion of gaze time in the key and associated areas of passing intuition ($|R| > 0.50$, $Ps < 0.01$). The tracking accuracy of the

**Table 3  Correlation between MOT accuracy and intuitive decision AOI fixation time proportion.**

| MOT | Correlation | Expert | | | Novice | | |
|---|---|---|---|---|---|---|---|
| | | Key | Associated | Key | Associated | Key | Associated |
| 2 | R | −0.368 | −0.274 | −0.004 | −0.325 | −0.106 | −0.285 |
| | P | 0.077 | 0.195 | 0.984 | 0.121 | 0.623 | 0.177 |
| 3 | R | −0.585** | −0.689** | 0.059 | −0.570** | −0.121 | −0.478* |
| | P | 0.003 | <0.001 | 0.786 | 0.004 | 0.572 | 0.018 |
| 4 | R | −0.658** | −0.637** | −0.259 | −0.464* | −0.427* | −0.546** |
| | P | <0.001 | <0.001 | 0.221 | 0.022 | 0.038 | 0.006 |
| 5 | R | −0.444* | −0.509* | −0.253 | −0.279 | −0.057 | −0.551** |
| | P | 0.030 | 0.011 | 0.233 | 0.187 | 0.791 | 0.005 |
| 6 | R | −0.423* | −0.433* | 0.159 | −0.378 | −0.384 | −0.379 |
| | P | 0.039 | 0.035 | 0.459 | 0.069 | 0.064 | 0.068 |
| 5°/s | R | −0.113 | −0.312 | 0.391 | −0.038 | 0.006 | −0.146 |
| | P | 0.601 | 0.138 | 0.059 | 0.858 | 0.977 | 0.496 |
| 10°/s | R | −0.511* | −0.637** | −0.232 | −0.524** | −0.153 | −0.468* |
| | P | 0.011 | <0.001 | 0.274 | 0.009 | 0.474 | 0.021 |
| 15°/s | R | −0.165 | −0.277 | 0.176 | −0.358 | −0.352 | −0.205 |
| | P | 0.441 | 0.189 | 0.410 | 0.086 | 0.091 | 0.337 |

Notes.
*$P < 0.05$.
**$P < 0.01$.

novice group was highly negatively correlated with the proportion of gaze time in the unrelated area of passing intuition ($R = -0.546$, $P = 0.006$).

When the number of MOT targets was five, the tracking accuracy of the expert group showed a highly negative correlation with the proportion of gaze time in the intuitive correlation zone of passing ($R = -0.509$, $P = 0.011$). For the novice group, tracking accuracy was highly negatively correlated with the proportion of gaze time in the unrelated area of passing intuition ($R = -0.551$, $P = 0.005$).

At an MOT target speed of 10°/s, the tracking accuracy of the expert group was highly negatively correlated with the proportion of gaze time in the key and associated areas of passing intuition $|R| > 0.50$, $Ps < 0.05$). In the novice group, the tracking accuracy was also highly negatively correlated with the proportion of gaze time in the key area of passing intuition ($R = -0.524$, $P = 0.009$). The correlation between MOT accuracy and the proportion of gaze time in areas unrelated to intuitive decision-making was lower for the expert group compared to the novice group. See Table 3.

## Correlation between MOT accuracy and the proportion of fixation time in cognitive decision-making interest areas

The results indicated that when the number of MOT targets was two, tracking accuracy in the expert group was highly negatively correlated with the proportion of gaze time in the cognitive key area of passing ($R = -0.540$, $P = 0.006$).

When the number of MOT targets was three, tracking accuracy in the expert group was highly negatively correlated with the proportion of gaze time in both the cognitive

key area of passing ($R = -0.699$, $P < 0.001$) and the cognitive association area of passing ($R = -0.772$, $P < 0.001$). In contrast, the tracking accuracy in the novice group was highly negatively correlated with the proportion of gaze time in both the critical and irrelevant areas of passing cognition ($R > 0.50$, $P < 0.001$).

When the number of MOT targets was four, tracking accuracy in the expert group was highly negatively correlated with the proportion of gaze time in both the key and associated areas of passing cognition ($R > 0.70$, $P < 0.001$). Similarly, tracking accuracy in the novice group was highly negatively correlated with the proportion of gaze time in both the critical and irrelevant areas of passing cognition ($R > 0.50$, $P < 0.001$).

When the number of MOT targets was five, the expert group showed a highly negative correlation between tracking accuracy and the proportion of gaze time in both the cognitive key area of passing ($R = -0.698$, $P < 0.001$) and the cognitive association area of passing ($R = -0.753$, $P < 0.001$). In the novice group, tracking accuracy was highly negatively correlated with the proportion of gaze time in both the critical and irrelevant areas of passing cognition ($|R| > 0.50$, $Ps < 0.001$).

When the number of MOT targets was six, tracking accuracy in the expert group was highly negatively correlated with the proportion of gaze time in both the key and associated areas of passing cognition ($|R| > 0.50$, $Ps < 0.001$). In the novice group, tracking accuracy was highly negatively correlated with the proportion of gaze time in the cognitive association zone of passing ($R = -0.523$, $P = 0.009$).

At an MOT target speed of 10°/s, tracking accuracy in the expert group was highly negatively correlated with the proportion of gaze time in both the key cognitive area of passing ($R = -0.578$, $P = 0.003$) and the cognitive association area of passing ($R = -0.831$, $P < 0.001$). In contrast, the tracking accuracy in the novice group was highly negatively correlated with the proportion of gaze time in both the critical and irrelevant areas of passing cognition ($|R| > 0.70$, $Ps < 0.001$).

The correlation between MOT accuracy in the expert group and the proportion of gaze time in the cognitive decision-making irrelevant area was lower than that observed in the novice group. See Table 4.

## Correlation between MOT accuracy and intuitive decision area of interest gaze frequency ratio

The results indicated that when the number of MOT targets was three, tracking accuracy in the expert group was highly negatively correlated with the proportion of fixations in both the key and associated areas of passing intuition ($|R| > 0.50$, $Ps < 0.01$). In contrast, tracking accuracy in the novice group was highly negatively correlated with the proportion of fixation times in the key area of passing intuition ($R = -0.500$, $P = 0.013$).

When the number of MOT targets was four, tracking accuracy in the expert group was highly negatively correlated with the proportion of fixations in the key area of passing intuition ($R = -0.610$, $P = 0.002$), and with the proportion of fixations in the associated area of passing intuition ($R = -0.724$, $P < 0.001$). Similarly, the tracking accuracy in the novice group was highly negatively correlated with the proportion of fixation times in the key area of passing intuition ($R = -0.628$, $P = 0.001$).
**Table 4  Correlation between MOT accuracy and the proportion of fixation time in cognitive decision-making AOI.**

| MOT | Correlation | Expert | | | Novice | | |
|---|---|---|---|---|---|---|---|
| | | Key | Associated | Associated | Key | Associated | Associated |
| 2 | R | −0.540** | −0.463* | −0.262 | −0.318 | −0.272 | −0.273 |
| | P | 0.006 | 0.023 | 0.217 | 0.130 | 0.198 | 0.196 |
| 3 | R | −0.699** | −0.772** | −0.466* | −0.697** | −0.417* | −0.693** |
| | P | <0.001 | <0.001 | 0.022 | <0.001 | 0.043 | <0.001 |
| 4 | R | −0.828** | −0.869** | −0.520** | −0.677** | −0.467* | −0.696** |
| | P | <0.001 | <0.001 | 0.009 | <0.001 | 0.021 | <0.001 |
| 5 | R | −0.698** | −0.753** | −0.486* | −0.666** | −0.131 | −0.665** |
| | P | <0.001 | <0.001 | 0.016 | <0.001 | 0.542 | <0.001 |
| 6 | R | −0.538** | −0.541** | −0.393 | −0.451* | −0.523** | −0.479* |
| | P | 0.007 | 0.006 | 0.058 | 0.027 | 0.009 | 0.018 |
| 5°/s | R | −0.081 | −0.212 | −0.285 | −0.181 | −0.085 | −0.262 |
| | P | 0.707 | 0.319 | 0.178 | 0.397 | 0.693 | 0.216 |
| 10°/s | R | −0.578** | −0.831** | −0.422* | −0.733** | −0.443* | −0.726** |
| | P | 0.003 | <0.001 | 0.040 | <0.001 | 0.030 | <0.001 |
| 15°/s | R | −0.128 | −0.277 | 0.081 | −0.342 | −0.385 | −0.331 |
| | P | 0.552 | 0.191 | 0.706 | 0.102 | 0.063 | 0.114 |

**Notes.**
 *$P < 0.05$.
 **$P < 0.01$.

When the number of MOT targets was five, tracking accuracy in the expert group was highly negatively correlated with the proportion of fixation times in the intuitive association area of passing ($R = -0.611$, $P = 0.002$). In the novice group, tracking accuracy was highly negatively correlated with the proportion of fixations in the area unrelated to passing intuition ($R = -0.528$, $P = 0.008$).

At an MOT target speed of 10°/s, tracking accuracy in the expert group was highly negatively correlated with the proportion of fixations in both the key and associated areas of passing intuition ($|R| > 0.50$, $Ps < 0.01$). In the novice group, tracking accuracy was highly negatively correlated with the proportion of gaze counts in both the key and irrelevant areas of passing intuition ($|R| > 0.50$, $Ps < 0.01$). See Table 5.

## Correlation between MOT accuracy and the proportion of fixations in cognitive decision areas of interest

The results indicated that when the number of MOT targets was three, tracking accuracy in the expert group was highly negatively correlated with the proportion of fixations in both the key and associated areas of passing cognition ($|R| > 0.50$, $Ps < 0.001$). In the novice group, tracking accuracy was highly negatively correlated with the proportion of fixation times in the key cognitive area of passing ($R = -0.719$, $P < 0.001$), and also highly negatively correlated with the proportion of fixation times in the unrelated cognitive area of passing ($R = -0.566$, $P = 0.004$).

When the number of MOT targets was four, tracking accuracy in the expert group was highly negatively correlated with the proportion of fixations in the cognitive association

**Table 5 Correlation between MOT accuracy and intuitive decision AOI fixation frequency.**

| MOT | Correlation | Expert | | | Novice | | |
|---|---|---|---|---|---|---|---|
| | | Key | Associated | Key | Associated | Key | Associated |
| 2 | R | −0.308 | −0.382 | 0.040 | −0.134 | −0.086 | −0.314 |
| | P | 0.143 | 0.065 | 0.851 | 0.534 | 0.688 | 0.135 |
| 3 | R | −0.613** | −0.661** | −0.190 | −0.500* | −0.098 | −0.396 |
| | P | 0.001 | <0.001 | 0.373 | 0.013 | 0.647 | 0.055 |
| 4 | R | −0.610** | −0.724** | −0.022 | −0.628** | −0.367 | −0.461* |
| | P | 0.002 | <0.001 | 0.921 | 0.001 | 0.078 | 0.023 |
| 5 | R | −0.349 | −0.611** | −0.090 | −0.487* | 0.051 | −0.528** |
| | P | 0.094 | 0.002 | 0.674 | 0.016 | 0.815 | 0.008 |
| 6 | R | −0.287 | −0.303 | −0.279 | −0.435* | −0.475* | −0.426* |
| | P | 0.173 | 0.151 | 0.186 | 0.033 | 0.019 | 0.038 |
| 5°/s | R | −0.083 | −0.069 | −0.476* | −0.239 | −0.091 | −0.098 |
| | P | 0.700 | 0.748 | 0.019 | 0.261 | 0.673 | 0.648 |
| 10°/s | R | −0.529** | −0.621** | −0.043 | −0.562** | −0.163 | −0.529** |
| | P | 0.008 | 0.001 | 0.842 | 0.004 | 0.445 | 0.008 |
| 15°/s | R | −0.340 | −0.167 | 0.301 | −0.354 | −0.382 | −0.286 |
| | P | 0.104 | 0.434 | 0.153 | 0.089 | 0.066 | 0.175 |

Notes.
*$P < 0.05$.
**$P < 0.01$.

area of passing ($R = -0.797$, $P < 0.001$). In the novice group, tracking accuracy was highly negatively correlated with the proportion of fixation times in the key cognitive area of passing ($R = -0.714$, $P < 0.001$), and also highly negatively correlated with the proportion of fixation times in the unrelated cognitive area of passing ($R = -0.602$, $P = 0.002$).

When the number of MOT targets was five, tracking accuracy in the expert group was highly negatively correlated with the proportion of fixations in the cognitive association area of passing ($R = -0.664$, $P < 0.001$). In the novice group, tracking accuracy was highly negatively correlated with the proportion of fixation times in the cognitive key area of passing ($R = -0.593$, $P = 0.002$).

When the number of MOT targets was six, tracking accuracy in the expert group was highly negatively correlated with the proportion of fixations in the cognitive association area of passing ($R = -0.501$, $P = 0.013$).

At an MOT target speed of 10°/s, tracking accuracy in the expert group was highly negatively correlated with the proportion of fixations in the cognitive association area of passing ($R = -0.653$, $P < 0.001$). In the novice group, tracking accuracy was highly negatively correlated with the proportion of fixation times in the key cognitive area of passing ($R = -0.715$, $P < 0.001$).

The correlation between MOT accuracy in the expert group and the proportion of gaze frequency in the cognitive decision-making irrelevant area was lower than that in the novice group. See Table 6.

Table 6 Correlation between MOT accuracy and cognitive decision-making AOI fixation frequency proportion.

| MOT | Correlation | Expert | | | Novice | | |
|---|---|---|---|---|---|---|---|
| | | Key | Associated | Key | Associated | Key | Associated |
| 2 | R | −0.161 | −0.432[*] | −0.007 | −0.317 | −0.371 | −0.277 |
| | P | 0.452 | 0.035 | 0.975 | 0.131 | 0.074 | 0.191 |
| 3 | R | −0.655[**] | −0.696[**] | −0.439[*] | −0.719[**] | −0.369 | −0.566[**] |
| | P | <0.001 | <0.001 | 0.032 | <0.001 | 0.076 | 0.004 |
| 4 | R | −0.449[*] | −0.797[**] | −0.114 | −0.714[**] | −0.363 | −0.602[**] |
| | P | 0.028 | <0.001 | 0.597 | <0.001 | 0.082 | 0.002 |
| 5 | R | −0.423[*] | −0.664[**] | −0.055 | −0.593[**] | −0.174 | −0.444[*] |
| | P | 0.039 | <0.001 | 0.798 | 0.002 | 0.417 | 0.030 |
| 6 | R | −0.408[*] | −0.501[*] | −0.067 | −0.453[*] | −0.250 | −0.297 |
| | P | 0.048 | 0.013 | 0.756 | 0.026 | 0.239 | 0.159 |
| 5°/s | R | −0.128 | −0.135 | −0.110 | −0.242 | −0.084 | 0.059 |
| | P | 0.552 | 0.531 | 0.609 | 0.255 | 0.696 | 0.785 |
| 10°/s | R | −0.476[*] | −0.653[**] | −0.405 | −0.715[**] | −0.315 | −0.433[**] |
| | P | 0.019 | <0.001 | 0.050 | <0.001 | 0.133 | 0.035 |
| 15°/s | R | −0.287 | −0.124 | −0.345 | −0.367 | −0.168 | −0.408[*] |
| | P | 0.174 | 0.563 | 0.099 | 0.078 | 0.432 | 0.048 |

Notes.
[*] $P < 0.05$.
[**] $P < 0.01$.

## DISCUSSION

In basketball games, players must not only monitor their teammates' movements but also continuously observe the defensive actions of their opponents. This scenario is akin to MOT tasks, which require players to track target changes over time. Through extensive training and competition, elite basketball players develop the ability to process visual information rapidly, facilitating more effective decision-making. This study found that the MOT accuracy of both expert and novice groups was negatively correlated with the average fixation time and average fixation frequency. When the number of targets ranged from three to six and the target speed was 10°/s, there was a strong correlation between MOT accuracy and eye movement indicators in sports decision-making tasks for both expert and novice players.

The correlation between the MOT accuracy of the expert group and the average fixation time in passing intuition and passing cognition was higher than that of the novice group. Conversely, the correlation between the proportion of fixation time in areas unrelated to intuitive decision-making was lower in the expert group than in the novice group. Similarly, the correlation between the proportion of fixation times in areas unrelated to cognitive decision-making was lower in the expert group. Previous research by the author has confirmed a relationship between MOT performance and sports decision-making performance. Specifically, the accuracy of MOT with four–five targets was positively correlated with passing decisions. Moreover, the correlation between MOT accuracy and

sports decision-making was more significant in expert players. Excessive tracking targets (greater than 6) can disrupt players' decision-making abilities (*Gou & Li, 2023*).

Fanghui Qiu's research shows that players experience decreased activation and functional connectivity of the dorsal attention network when performing non-sport-specific cognitive tasks, extending the neural efficiency hypothesis, which is typically applied in sports training, to general cognitive domains. The neural efficiency hypothesis posits that experts exhibit more efficient cortical functions, leading to improved cognitive performance and reduced energy expenditure (*Qiu, 2019*). This study provides a deeper exploration of this concept from the perspective of visual processing in sports decision-making. *Zhang, Zhuang & Chen (2009)* suggest that players' predictive ability is closely related to their visual search capacity. Furthermore, Peng Jin's research indicates that expert players display shorter fixation times and fewer fixations during competitions, focusing more on key areas. The associated areas of interest play a crucial role in action prediction, and the stimulus images in these areas suggest the action' s trend. However, the real challenge lies in determining the associated areas for the subsequent steps (*Jin et al., 2020b*).

Yaqiong Cao's research underscores the importance of selective attention in players, enabling them to make comprehensive judgments about the game situation and focus on critical areas and suitable teammates. This allows them to take effective actions after making sound decisions (*Cao & Li, 2010*). These findings highlight a significant phenomenon: the limitation of attentional resources in cognitive processes. When individuals allocate more attention to irrelevant areas, it diminishes their ability to focus on crucial areas, which often contain important clues for decision-making.

Passing is a fundamental and critical technique in basketball, directly influencing offensive tactics and game outcomes (*Zheng, 2012*). It is not only a core skill but also a key element in executing various strategies. A well-timed pass can often alter the course of a game and even determine the winner. Research has shown that precise passing requires players to make quick and accurate judgments within a limited timeframe. Such rapid decisions are dependent on acquiring visual stimuli promptly, making visual acuity essential for performance (*Niu, 2019*).

This study found that the MOT accuracy of the expert group was higher than that of the novice group, particularly with respect to the average fixation time for passing intuition and passing cognition. The ability to track multiple targets was positively correlated with visual processing related to passing decisions. Professional players, due to extensive training and competition experience, are better equipped to process information quickly compared to novices. The visual skills developed through such training enable expert players to make more rational decisions in competitive situations. The results of this study support the hypothesis that players with strong visual information processing abilities perform better in sports decision-making tasks. However, sports decision-making is influenced by various factors. A non-adaptive design may not capture individual variation in tracking ability as effectively as staircase/adaptive methods. Given that ecological validity is one of the central concerns in applied perceptual-cognitive research, In the future, we need to further validate it in real basketball games. The aim of this study was to demonstrate that excellent visual attention is crucial for basketball players during competitions and is one of the essential

psychological attributes for players. The study found that the multiple object tracking ability of basketball players was negatively correlated with fixation time and frequency in sports decision-making tasks, while positively correlating with fixation time in passing decisions, providing valuable insights for sports decision-making training.

## CONCLUSIONS

(1) The accuracy of MOT in both expert and novice groups is negatively correlated with the average fixation time and average fixation frequency. The correlation between the MOT accuracy of the expert group and the average fixation time for passing intuition and passing cognition is higher than that of the novice group. (2) When tracking three–six targets at a speed of 10°/s, there is a strong correlation between the MOT accuracy of the expert and novice groups and the eye movement indicators in the sports decision-making task. (3) The correlation between the MOT accuracy in the expert group and the proportion of fixation time in areas unrelated to intuitive decision-making is lower than that in the novice group. Similarly, the correlation with the proportion of fixation times in areas unrelated to cognitive decision-making is also lower in the expert group.

### Funding
The present study was funded by the Northwest Normal University Young Teachers' Research Ability Improvement Project (No. NWNU-SKQN2023-35), the Gansu Province Sports Scientific Research and Decision-making Consultation Project (No. 2023-C-20), and the National Social Science Foundation of China (No. 22BTY055). The funders had no role in study design, data collection and analysis, decision to publish, or preparation of the manuscript.

### Grant Disclosures
The following grant information was disclosed by the authors:
The Northwest Normal University Young Teachers' Research Ability Improvement Project: No. NWNU-SKQN2023-35.
The Gansu Province Sports Scientific Research and Decision-making Consultation Project: No. 2023-C-20.
The National Social Science Foundation of China: No. 22BTY055.

### Competing Interests
The authors declare there are no competing interests.

### Author Contributions
- Qifeng Gou conceived and designed the experiments, performed the experiments, analyzed the data, prepared figures and/or tables, authored or reviewed drafts of the article, and approved the final draft.
- Sunnan Li conceived and designed the experiments, prepared figures and/or tables, authored or reviewed drafts of the article, and approved the final draft.

## Human Ethics

The following information was supplied relating to ethical approvals (*i.e.*, approving body and any reference numbers):

This work was approved by the regional ethics committee of the Northwest Normal University (No. NWNU-20230301).

## Data Availability

The raw measurements are available in the Supplementary Files.

## Supplemental Information

Supplemental information for this article can be found online at http://dx.doi.org/10.7717/peerj.19984#supplemental-information.

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
