# Peer review of "Study on the correlation between multiple object tracking ability and eye- tracking characteristics in sports decision making among basketball players"

_PeerJ, doi:10.7717/peerj.19984_

## Round 0.1 · original submission · Major Revisions

The manuscript has been reviewed by two experts in the field. Revisions are
necessary before the manuscript is suitable for publication.

·

Basic reporting

Expand the introduction to better define how this study builds upon existing work
Implications for practice can be better highlighted; provide tangible takeaways for readers to practice and improve their sport performance.

Experimental design

Provide justification on why female players were selected
Discussion on potential gender-specific effects or limitations
The decision-making can be highlighted with clearer operational definations
Some sentences are lengthy and could be more concise
Additional references on eye-tracking methodologies in sports could enhance the background

Validity of the findings

The manuscript employs appropriate statistical tests (Pearson correlation) but lacks discussion on effect sizes and confidence intervals.
Factors such as prior eye-tracking experience, fatigue, or visual acuity could have influenced results and should be discussed as potential confounds.
Correlation coefficients are useful, but adding effect sizes could strengthen the findings.

Additional comments

NA

Reviewer 2 ·

Basic reporting

The manuscript is generally well written and logically structured. The language is clear and professional, with appropriate use of terminology. The introduction provides relevant theoretical background, referencing both classical and recent literature in the field of perceptual-cognitive research and sports decision-making. The structure of the article, including the organization of tables and figures, adheres to the journal's standards.

However, several aspects require clarification:
• Definition of terms: The concepts of “passing intuition” and “passing cognition” appear frequently, but they are not clearly defined. As these are key variables in the study, I recommend providing precise operational definitions supported by relevant literature.
• Justification of hypotheses: The research hypotheses are implied rather than clearly formulated. They should be explicitly presented and better anchored in previous empirical studies to strengthen the manuscript’s theoretical framework.
• Presentation of correlation coefficients (R) in the results section: In the text, the authors describe the strength of correlations using absolute values (|R|), while in other parts and tables they present actual R values with directional signs. Although this is acceptable, I recommend unifying the presentation of these values or clearly explaining the adopted convention (e.g., in a footnote or a brief methodological note) to avoid interpretational ambiguities.
• The method of eye movement recording (monocular vs. binocular) is not specified. Since the Eyelink Portable Duo system supports both, it would be useful to indicate the chosen mode, as it affects the interpretation of fixation and saccade results and enhances replicability.
• The limitations section should be expanded. For example: the expert group consisted only of players from one university league (Chinese University Basketball League), which limits the diversity of skill levels and the generalizability of the findings. Additionally, the ecological validity of the study is limited due to the laboratory conditions and lack of motor responses.

Experimental design

The study addresses a relevant topic within the scope of the journal – examining the relationship between multiple object tracking (MOT) ability and eye movement parameters during decision-making in basketball. The use of the expert–novice paradigm is appropriate and consistent with previous literature.

However, several methodological issues should be noted:
• Data distribution: No information is provided regarding tests for normality of data distribution. Given the use of Pearson correlations, such tests should be reported. If normality is violated, non-parametric alternatives should be considered.
• Lack of adaptive method: The MOT task used fixed numbers of objects and speeds (2–6 objects, 5–15°/s), without applying an adaptive (staircase) method, which would better capture individual capabilities. The choice of fixed difficulty levels should be justified or listed as a limitation.
• MOT measurement: MOT ability was assessed solely based on accuracy (percentage of correctly identified targets). Including other metrics (e.g., response time, measurement reliability, confidence ratings) would improve the robustness of the findings.
• Ecological validity: The use of first-person video enhances realism, but the absence of actual motor decisions limits the ecological validity and applicability of results to real-game performance.

Validity of the findings

Statistical analyses were conducted thoroughly, and the results are clearly presented and appropriately related to the studied variables.

Nonetheless, the following concerns should be noted:
• Lack of control for confounding variables: The study does not account for potential factors such as visual acuity, cognitive capacity, fatigue, or motivation, which may influence both MOT and decision-making performance.
• Correlational nature: As the study is correlational, causal inferences should be avoided. Although the authors generally acknowledge this, some parts of the discussion could more clearly state that the results do not confirm causal directions. For example, it is unclear whether better MOT leads to better decisions, or if game experience improves tracking ability.
• Risk of Type I error: Due to the large number of correlation tests performed, there is an increased risk of Type I error. I recommend applying corrections for multiple comparisons (e.g., Bonferroni or FDR), or at least addressing this issue as a limitation in the discussion.

Additional comments

This study contributes to the growing field of perceptual-cognitive research in sport. The integration of multiple object tracking and eye-tracking measures in the context of decision-making among basketball players is both relevant and timely.

To further improve the quality and clarity of the manuscript, I suggest:
• Clearly outlining the main hypotheses at the end of the introduction, ensuring they are explicitly linked to the literature.
• Expanding the discussion section to include limitations related to sample representativeness, lack of motor execution, and the laboratory nature of the study environment.
• Highlighting possible practical implications for perceptual-cognitive training programs in team sports.
• Providing details on how the dependent variables were operationalized (e.g., what thresholds were used to define correct or incorrect decisions).
• Ensuring consistent terminology and reporting conventions throughout the manuscript.

---

## Round 0.2 · Minor Revisions

Minor revisions are necessary before the manuscript is suitable for publication.

**PeerJ Staff Note**: Please ensure that all review, editorial, and staff comments are addressed in a response letter and that any edits or clarifications mentioned in the letter are also inserted into the revised manuscript where appropriate.

Reviewer 2 ·

Basic reporting

I would like to thank the authors for the work put into revising the manuscript. The updated version shows careful consideration of many earlier suggestions, and I appreciate the improved clarity resulting from the inclusion of explicit hypotheses, methodological clarifications, and an expanded discussion of limitations and practical implications.
That said, a few aspects still require refinement to fully meet the journal’s expectations of methodological transparency and scientific rigor. Please find below my detailed feedback on the remaining points.
1. The authors have added a reference to Wang (2005) and clarified the temporal thresholds for “passing intuition” (≤ 0.6 s) and “passing cognition” (≤ 1.2 s) in the context of decision-making. While this clarification is helpful, I suggest that the operational definitions could be expanded further. Specifically, the current explanation focuses only on response time, without elaborating on the underlying cognitive mechanisms, attentional demands, or decision strategies that distinguish these two modes of processing. Additional references and a clearer theoretical grounding would strengthen the conceptual framework and allow readers to better interpret the results.
2. I appreciate the authors’ explanation and literature citations regarding the interpretation of correlation coefficients. However, I respectfully note that the current manuscript inconsistently reports correlation values, sometimes using the absolute value |R| and at other times the signed R. To enhance clarity and avoid ambiguity, especially when discussing directionality of effects, I recommend consistently reporting signed R values (e.g., r = –0.68) throughout both the results and tables, and using terms like positive or negative correlation accordingly. If absolute values are used in a summary, it would be helpful to explicitly state why and where this convention applies (e.g., in a footnote or figure legend).

Experimental design

3.In response to the reviewer’s comment, the authors state that the data follow a normal distribution, based on precedent in a previous study. However, for transparency and scientific completeness, it is typically expected that authors report how this assumption was verified, especially when using parametric statistics such as Pearson correlations. I suggest briefly noting in the Methods section which test for normality was used (e.g., Shapiro-Wilk), or alternatively including normality diagnostics in the supplementary materials.
4. The response points to earlier studies as justification for using a fixed set of object numbers and speeds. While this does align with precedent, I would still encourage the authors to clearly acknowledge in the limitations section that a non-adaptive design may not capture individual variation in tracking ability as effectively as staircase/adaptive methods.
5. The authors note the issue of ecological validity as a direction for future research, which is appreciated. However, given that ecological validity is one of the central concerns in applied perceptual-cognitive research, I kindly suggest that this limitation be more clearly addressed in the discussion. A brief acknowledgment of how the absence of actual motor responses may affect attentional strategies or decision-making dynamics would help strengthen the ecological interpretation of the findings

Validity of the findings

6. The authors state that visual acuity, cognitive capacity, fatigue, or motivation were not the focus of the study. While this is acknowledged, I would still recommend explicitly addressing the absence of these controls in the limitations section. Not accounting for such factors may introduce uncontrolled variance that could influence both MOT performance and decision-making outcomes. A brief acknowledgment that these variables were not measured or controlled would improve methodological transparency and help contextualize the findings more accurately. It would also be beneficial to support this point with references to prior research demonstrating the potential influence of such confounding variables on perceptual-cognitive performance.
7. The authors refer to Jin et al. (2020b) to justify the statistical approach, but they do not address the reviewer’s concern regarding the large number of pairwise correlation tests conducted. While correction procedures such as Bonferroni or FDR are not always mandatory, it is important to acknowledge that performing many individual tests increases the risk of Type I error. Including a brief statement to this effect in the statistical analysis section or the discussion would help to contextualize the findings and encourage cautious interpretation of marginally significant results. Ideally, this point should also be supported by references to methodological literature that highlights the implications of multiple comparisons in correlational research.

---

## Round 0.3 · accepted · Accept

I have now had the opportunity to read your revised manuscript, and your responses to the reviewers’ comments. I believe that you have addressed the concerns raised, and I am happy to accept your manuscript.

Reviewer 2 ·

Basic reporting

no comment

Experimental design

no comment

Validity of the findings

no comment

Additional comments

Most of the reviewer’s comments have been addressed accurately and in accordance with scientific standards. A few responses are only partially elaborated, but they do not significantly affect the quality of the work or the interpretation of the results.